# Gut Microbiome–Colorectal Cancer Relationship

**DOI:** 10.3390/microorganisms12030484

**Published:** 2024-02-27

**Authors:** Devvrat Yadav, Chiranjeevi Sainatham, Evgenii Filippov, Sai Gautham Kanagala, Syed Murtaza Ishaq, Thejus Jayakrishnan

**Affiliations:** 1Department of Internal Medicine, Sinai Hospital of Baltimore, 2401 W Belvedere Ave, Baltimore, MD 21215, USAefilippov@lifebridgehealth.org (E.F.); sishaq@lifebridgehealth.org (S.M.I.); 2Department of Internal Medicine, NYC Health + Hospital/Metropolitan, New York, NY 10029, USA; 3Division of Hematology and Oncology, Cleveland Clinic, Cleveland, OH 44195, USA

**Keywords:** gut microbiome, colon cancer, colorectal cancer, chemotherapy, immunotherapy, immune checkpoint inhibitors, fecal microbiota transplantation

## Abstract

Traditionally, the role of gut dysbiosis was thought to be limited to pathologies like *Clostridioides* difficile infection, but studies have shown its role in other intestinal and extraintestinal pathologies. Similarly, recent studies have surfaced showing the strong potential role of the gut microbiome in colorectal cancer, which was traditionally attributed mainly to sporadic or germline mutations. Given that it is the third most common cancer and the second most common cause of cancer-related mortality, 78 grants totaling more than USD 28 million have been granted to improve colon cancer management since 2019. Concerted efforts by several of these studies have identified specific bacterial consortia inducing a proinflammatory environment and promoting genotoxin production, causing the induction or progression of colorectal cancer. In addition, changes in the gut microbiome have also been shown to alter the response to cancer chemotherapy and immunotherapy, thus changing cancer prognosis. Certain bacteria have been identified as biomarkers to predict the efficacy of antineoplastic medications. Given these discoveries, efforts have been made to alter the gut microbiome to promote a favorable diversity to improve cancer progression and the response to therapy. In this review, we expand on the gut microbiome, its association with colorectal cancer, and antineoplastic medications. We also discuss the evolving paradigm of fecal microbiota transplantation in the context of colorectal cancer management.

## 1. Introduction

The human microbiome is a collection of microbial communities that constantly change as specific communities of microbes occupy anatomical niches within the human body. This colonization process starts soon after birth upon exposure to the vaginal microbiota. Infants continue to be introduced to new flora through routine activities with other humans, including feeding and playing, resulting in the establishment of the microbiome on the skin, gut, and mucosal surfaces. The introduction and reintroduction of flora continue throughout life from our routine interactions.

The microbiome, defined as the number of microorganisms with their genetic material, often significantly impacts human health. It is very different from the microbiota, which refers to the microbial population in different ecosystems in the body [1,2]. These microbiotas carry out many complex biochemical and metabolic functions [3]. The indigenous microbiota can also modify epithelial responses and systemic responses, such as the development and activity of the immune system with alterations in the antitumor responses. Any disturbance to the symbiosis between mammalian hosts and their microbial partners can lead to various diseases, including direct effects on the development and progression of colorectal cancer (CRC). As the third most common cancer in the US, with an estimated burden of 153,020 cases per 100,000 person-years in 2023, and the second most common cause of cancer death, with estimated 52,550 deaths per 100,000 person years, colorectal cancer adds a significant morbidity and healthcare burden to the US [4]. Meanwhile, it has the second highest treatment cost for any cancer, accounting for 12.6% of all cancer treatment costs and equating to a total of USD 24.3 billion annually in 2020 [5]. Hence, there have been ongoing efforts in the research and development of new therapeutic options to treat CRC and, if possible, identify the biomarkers required for its early detection. Manipulating the gut microbiome by means of fecal transplantation is emerging as one of the many novel therapeutic options to treat CRC and improve its response to chemotherapy and immunotherapy. This review focuses on the pivotal connection between the gut microbiome and colorectal cancer, examining its potential for modulating tumorigenesis and presenting treatment strategies [6].

## 2. Gut Microbiome’s Essential Role in Cancer Prevention

A healthy microbiome is essential in maintaining homeostasis, and certain species have shown their direct effect in preventing cancer. The most extensively studied protective microorganisms are *Bifidobacterium* and *Lactobacillus* spp, which have demonstrated anti-cancer properties by employing various mechanisms. These mechanisms encompass their influence on cellular proliferation and apoptosis, their regulation of host immunity, and their neutralization of carcinogenic toxins and xenobiotics [7,8]. However, other colonic bacteria can also exhibit anti-inflammatory and anti-carcinogenic effects. For instance, ingested dietary fibers undergo fermentation by butyrate-producing microbes (*Ruminococcus*, *Clostridium*, *Eubacterium*, and *Faecalibacterium*), producing short-chain fatty acids (SCFA). SCFA interacts with G protein-coupled receptors, inhibiting histone deacetylase in the colonic epithelium and immune cells. This interaction results in histone hyperacetylation, increasing T-reg numbers and promoting the production of the anti-inflammatory cytokines IL-10 and TGF-beta. Simultaneously, SCFA downregulates the production of the proinflammatory cytokines IL-6 and IL-12 in colonic macrophages, creating an anti-inflammatory microenvironment [9,10]. Moreover, by acting as a histone deacetylase inhibitor, butyrate also promotes the expression of tumor suppressor proteins such as FAS, p21, and p27 [11]. Some bacteria are also known for maintaining the barrier function. *Lactobacillus acidophilus*, *Bifidobacteria bifidum*, and *Bifidobacteria infantum* promote the increased expression of mucin 2 (MUC2), zonula occludens (ZO-1), and occludin, which are an essential part of intestinal epithelial barrier integrity and have shown an association with decreased colorectal tumor incidence and volume [12]. *Bifidobacterium longum* and Bacillus subtilis have been shown to downregulate proinflammatory cytokines (e.g., IL-17, IL-23, and TNF-α) and upregulate tight junction (TJ) proteins (e.g., claudin-1, occludin, and ZO-1) in colitis mouse models, preserving intestinal barrier function [13,14,15]. One actively investigated mechanism through which the microbiome protects against CRC involves the downregulation of IL-22 by the BL 23 strain of *Lactobacillus casei* [16]. Even though the species mentioned above are among the many that promote gut homeostasis, other species that tip the balance towards oncogenesis have also been identified, as detailed in the next section.

## 3. Abnormal Gut Microbiome in Patients with Colorectal Cancer

### 3.1. Streptococcus Bovis

*Streptococcus bovis* (*S. bovis*) is a Gram-positive bacterium, appearing as cocci in chains, with a facultative anaerobic lifestyle, and belongs to the family Streptococcaceae. It has been identified as one of the risk factors for colorectal cancer (CRC) [17,18,19]. *S. bovis* is typically found in the gastrointestinal tract. The occurrence of *S. bovis*-induced endocarditis or bacteremia was an early indication of its involvement in colorectal cancer [20]. The link between inflammation and colon carcinogenesis was further confirmed when the pro-inflammatory potential of *S. bovis* proteins and their carcinogenic properties were observed. It has been discovered that *S. bovis* actively contributes to the development of CRC, possibly through an inflammation-based sequence of tumor development or propagation involving interleukin (IL)-1, cyclooxygenase-2 (COX-2), and IL-8 [21,22].

### 3.2. Fusobacteria

*Fusobacteria* are Gram-negative, non-spore-forming, spindle-shaped obligate anaerobes belonging to the family *Fusobacteriaceae*, and have recently been at the forefront of discussions regarding the microbiome and tumor-associated pathogens [23]. Studies have shown that they originate from the oral microflora, with enteral transmission being the predominant route for CRC-tissue colonization by *F. nucleatum* [24,25]. These bacteria produce a unique protein called Fusobacterium adhesin A (FadA), which activates the β-catenin signaling pathway after binding to E-cadherin, a potent oncogenic stimulator [26,27]. Meanwhile, they also produce Fap2, a galactose adhesion hemagglutinin which mediates the colonization and invasion of CRC cells [28]. *F. nucleatum* promotes oncogenesis via various mechanisms, including promotion of a protumorigenic inflammatory milieu along with the inhibition of anticancer immune responses through genetic and epigenetic alterations [28,29,30,31,32,33,34,35]. In addition, it may also promote chemoresistance in CRC [36,37,38,39]. It has also been shown to be a potential marker for early detection, as well as prognostication and prediction of outcomes in CRC [40,41,42].

### 3.3. Enterococcus Fecalis

*Enterococcus faecalis* is a Gram-positive bacterium with a cocci shape from *Entercoccaceae* family, exhibiting a facultative anaerobic lifestyle. It is a gut commensal bacterium that produces superoxides through the autooxidation of membrane-associated demethylmenaquinone [43]. Infection with *E. faecalis* leads to superoxide-mediated DNA damage in intestinal epithelial cells. Studies have shown that the abundance of *E. faecalis* is significantly higher in CRC patients compared with healthy individuals [44,45,46]. Additionally, both in vitro and in vivo experiments have demonstrated that *E. faecalis* can produce hydroxyl radicals, which are highly mutagenic and can cause DNA breaks, point mutations, and protein–DNA crosslinking. These effects contribute to chromosomal instability and increase the risk of CRC.

### 3.4. Anaeroplasma

*Anaeroplasma* is a bacterial genus within the class Mollicutes, and its bacterial characteristics vary based on the specific genus. It survives in anaerobic environments and is typically found in the gastrointestinal and urogenital tract. Studies of its association with patients of CRC are conflicting. It is a fact that chronic colonic inflammation is a risk factor for CRC. In this context, studies have shown a relative higher abundance of *Anaeroplasma* in colitis mouse models with decrease in levels following healthy gut microbiome transfer [47]. Other studies have shown a higher abundance of *Anaeroplasma* in mutated APC gene mouse models than controls. The same study showed higher *Anaeroplasma* in older mice with a higher colonic tumor burden than younger mice with no tumors [48]. Another study showed conflicting results where chronic colitis mouse models treated with AOM (azoxymethane) had relatively lower *Anaeroplasma* abundance than the controls [49].

### 3.5. Flavobacteria

*Flavobacteria* are Gram-negative, rod-shaped bacteria belonging to the family Flavobacteriaceae, with studies showing conflicting evidence. One study showed a decreased concentration of this genus in patients with CRC with an underlying mechanism described as the destruction of mucosa-adherent microbiota in healthy individuals [50]. Another study demonstrated a relative abundance of flavobacteria in patients with colorectal adenomas [51]. Flavonoids are well-known for their anticancer properties, and their biodegradation, secondary to *Flavobacteria*, could be the primary cause of this pathology [52]. These conflicting results emphasize the possibility that single species do not reflect genus differences as a whole. This might lead to inconsistent results, as emphasized in previous studies although the variation could also reflect differences between adenomas and cancer [53].

### 3.6. Ruminococcaceae

The Ruminococcaceae family consists of Gram-positive, anaerobic bacteria with varying rod-shaped morphologies and plays an essential role in the fermentation of complex carbohydrates and the metabolism of dietary fibers. Even though these bacteria have shown increased prevalence in probiotics, the promotion of gut barrier integrity and the suppression of colonic carcinogenesis by production of secondary bile acids, studies have shown conflicting evidence [54,55,56]. One study identified the relative abundance of *Ruminococcus* in the colon of rats with underlying precancerous lesions induced by carcinogen 1,2-dimethyl hydrazine [57]. Another mouse study showed a significant increase in *Ruminococcaceae* in CRC group, with many other studies showing similar results [58,59,60]. In contrast to this, another study comparing normal, precancerous, and cancerous colonic tissue showed an abundance of *Ruminococcus* in normal tissues [61]. Another study showed decreased prevalence of *Ruminococcus gnavus* in mice treated with azoxymethane, which induces inflammation of the colon [49]. The same study further showed a negative correlation with tumor numbers, disease score, and inflammatory T cell subsets and a positive correlation with CD4+, FoxP3+, Tregs, and IL-10-producing T cells [49]. A recent study performing integrated analyses of differentially abundant bacterial groups (ASVs—amplicon sequence variants) of 1056 stool samples to identify biomarkers associated with adenoma and CRC showed *Ruminococcaceae UCG-005* as one of the two top-ranking biomarkers between controls and adenoma patients [62]. In the same study, the *Ruminococcus gnavus* group showed differential abundance between adenoma and cancer compared with healthy controls.

### 3.7. Acidovarax

*Acidovarax* are Gram-negative bacteria with variable shapes, exhibiting aerobic or anaerobic characteristics. They consist of acid-degrading bacteria from the family Comamonadaceae, promoting inflammation by metabolizing nitro-aromatic compounds and flagellar proteins that induce local inflammation [63]. This is hypothesized to be the potential reason behind the *Acidovarax*–adenoma association. Multiple studies analyzing these bacteria at the genus level have shown a relative abundance of *Acidovarax* in patients with adenoma vs. controls without adenoma [50,64].

### 3.8. Eubacteria

*Eubacteria* are Gram-positive anaerobic bacteria with a rod-shaped morphology, belonging to the family Eubacteriaceae. They are identified as bacteria with anti-inflammatory properties, with studies showing decreased anti-inflammatory effects in mice deficient in *Faecalibateria* and *Eubacteria* [65]. Another study specifically designed to look at candidate strains with anti-CRC activity showed *Eubacterium callanderi* to have antiproliferative properties against CRC cells by inducing apoptosis and cell death in a dose-dependent manner [66]. Further, the same study showed higher butyrate concentrations with the peri-tumoral injection of a cell-free supernatant of *Eubacteria* inhibiting tumor growth, further emphasizing their role in probiotic therapy to prevent CRC [66].

### 3.9. Bifidobacteria

*Bifidobacteria* are Gram-positive anaerobic rod-shaped bacteria belonging to the family Bifidobacteriaceae. They have recently garnered a lot of attention because of their beneficial effect on the gut microbiome. They are considered as a health-promoting gut microorganisms, especially beneficial in CRC prevention by means of the downregulation of anti-apoptotic and the upregulation of pro-apoptotic genes including BAD, Bxl-2, caspase-3, caspase-8, caspase-9 and Fas-R [67]. Another recent murine model study emphasized the oncoprotective effect of *Bifidobacterium* in the colon by means of the downregulation of the urea cycle [68]. Urea cycle activation is linked to microbial dysbiosis and excess urea could enter the macrophages, inhibiting the binding efficiency of p-STAT1 to the SAT1 promoter region, causing accumulation of polyamines and thus skewing macrophages to a pro-tumoral phenotype. These murine models, when treated with urea cycle inhibitors or Bifidobacterium-based probiotics, showed lower expression of Ki67 and CD206, which are markers of increased proliferation and macrophage immunosuppression, respectively. This emphasizes the role of *Bifidobacterium*-based probiotic supplements in mitigating CRC.

### 3.10. Others

Besides the abovementioned bacteria, there are many others that have shown associations with CRC. One study showed an increased prevalence of *Akkermansia* in longstanding colitis models [47]. Another study performed on humans analyzing the oral microbiome showed a decreased prevalence of *Prevotella* in the CRC vs. healthy group [69]. A few extensive metagenomic analyses of CRC datasets have shown numerous species to be present in the carcinoma-enriched environment, namely *Fusobacterium nucleatum*, *Parvimonas micra*, *Gemella morbiliorum*, *Peptostreptococcus* spp, *Solobacterium moorei*, *Clostridium symbiosum*, *Anaerococcus* spp, *Porphyromonas* spp, *Prevotella intermedia*, *Bacteroides fragilis*, *Streptococcus constellatus*, *Granuclicatella adiacens*, *Treponema denticola*, *Porphyromonas gingivalis* and *Tannerrella forsthica* [24,25,70]. Bacteria that showed increased prevalence in the controls included *Roseburia intestinalis*, *Gordonibacter pamelaeae* and *Bifidobacterium catenulatum* [24].

Expanding on the breadth of data available, studies even show changes in microbiome proportions in the colorectal adenoma vs. carcinoma populations, with various bacteria like *Butyricimonas synergistica*, *Agrobacterium larrymoorei*, *Bacteroides plebeius*, *Clostridium scindens*, *Lachnospiraceae bacterium* feline oral taxon 001, *Prevotella heparinolytica*, *Streptococcus mutans*, *Lachnospiraceae bacterium* 19gly4, and *Eubacterium hallii* showing the best performance in distinguishing colorectal adenoma from CRC populations [53]. Interestingly, other studies have also shown a differential abundance of bacteria in patients with CRC when comparing the right side versus the left side of the colon, with increased prevalence of *Haemophilus* and *Veilonella* in right-sided CRC patients vs. increased *Roseburia* and *Akkermansia* in left-sided CRC patients [71]. The role of microbiome has also been explored in the context of increasing incidence of young or early onset CRC. It appears that certain bacterial genera such as *Akkermansia* and *Bacteriodes* are differentially abundant in young individuals who develop CRC as opposed to several other genera (*Bacillus*, *Staphylococcus*, *Listeria*, *Enterococcus*, *Pseudomonas*, *Fusobacterium*, and *Escherichia/Shigella)* that are more abundant in CRC arising at the usual age [72]. Approaches such as plasma metabolomics analysis and machine learning are being used to further define the relationship between the altered microbiome in young individuals and CRC to identify microbiome-derived signatures for screening and therapy [73,74].

## 4. Biochemistry and Microbiome of Patients with Colorectal Cancer

Presently, two hypotheses have been formulated to describe the diverse mechanisms through which the gut microbiota can impact carcinogenesis in CRC.

### 4.1. Alpha Bug Model

The Alpha Bug Model hypothesis was initially formulated by Cynthia L. Sears and Drew M. Pardoll in 2011 [75]. It states that specific “Alpha-bugs” possess virulence factors capable of not only directly or indirectly initiating carcinogenesis but also displacing protective bacterial species, thereby reshaping the surrounding bacterial community [75]. The foundation for this hypothesis was formed by studying enterotoxigenic *Bacteroides fragilis* (ETBF), a bacterium that generates Bacteroides fragilis toxin (BFT). Research has demonstrated that the presence of ETBF in the intestines of the multiple intestinal neoplasia (Min) mice (a commonly employed murine model of CRC) led to the development of microscopic adenomas merely one-week post-colonization [76]. Since then, multiple studies have demonstrated the existence of numerous other gut bacteria (alpha-bugs) that can potentially induce a similar chain of events leading to cancer, including superoxide-producing *S. bovis*, *Enterococcus faecalis*, and *Escherichia coli* [77,78,79].

### 4.2. Driver–Passenger Hypothesis

The novel hypothesis based on the driver–passenger model was introduced by Harold Tjalsma, PhD, in 2012 [80]. This is an extended version of the alpha-bug model. According to this hypothesis, drivers (alpha-bugs) initiate CRC and cause changes in the tumor’s microenvironment, creating conditions suitable for colonization by opportunistic bacteria, known as ‘passengers.’ Passenger bacteria contribute to the progression of the disease and, throughout the illness, may even displace the driver pathogens. Passengers exert no influence on the initiation of tumorigenesis, but they may participate in the progression of the disease. Examples of passengers include *S. Gallolyticus*, *Fusobacterium*, and *Veillonella* spp. Furthermore, certain bacteria can fulfill dual roles as both passengers and drivers. For instance, *F. nucleatum* can trigger tumorigenesis and is abundant in patients with advanced CRC without being displaced. Supporting evidence for the driver–passenger model was provided through several additional empirical investigations [81]. Nevertheless, a clear differentiation categorizing a specific species as either a “driver” or a “passenger” in CRC continues to elude the definition. 

Regardless of the framework considered, the pathophysiology of colorectal cancer concerning the microbiome can be described through the three mechanisms described below: chronic inflammation, metabolism of dietary components, and generation of genotoxins.

### 4.3. Inflammation

Chronic inflammation and tumorigenesis are intricately interconnected, particularly in the context of CRC. Consequently, individuals with inflammatory bowel disease (IBD) exhibit a significantly heightened likelihood of developing CRC compared with the general population [82]. Moreover, the microbiota can also trigger inflammation, thereby contributing to the oncogenic process. For instance, as mentioned earlier, the Bacteroides fragilis toxin (BFT) produced by *B. fragilis* impacts local T-cell immunity by activating STAT3 and elevating Th17 and IL-17 levels. STAT3 is an oncogenic transcription factor, and when activated, it is translocated to the cell nucleus and increases the expression of such genes as Bcl-xL, Cyclin D1, and IL-6, thereby inhibiting apoptosis, promoting cellular proliferation and increasing the inflammatory response, therefore promoting abnormal cell line growths, [83]. On the other hand, an increased IL-17 level serves as a chemoattractant for myeloid-derived suppressor cells (MDSC). These cells secrete IL-1, IL-6, and TNF, further fueling chronic inflammation [76,84]. Moreover, IL-6 itself can activate STAT3, leading to a feed-forward loop where STAT3 further increases the production of IL-6 and other inflammatory mediators via the NF-kB-IL-6-Stat3 cascade [83]. Another example of chronic inflammatory changes induced by bacteria can be seen for *F. nucleatum*. This organism has been shown to increase nuclear factor-kappa B (NF-κB) expression [85,86]. NF-κB is a central immune response regulator and inflammation regulator that upregulates many chemokines (CXCL1, CXCL2, CXCL3) and cytokines (TNFα, IL-1β, IL-6, and IL-8). Both CXCL1 and CXCL2 recruit MDSCs to the tumor microenvironment, which again, in turn, creates a pro-inflammatory and carcinogenic environment. *F. nucleatum* also activates Wnt signaling pathways while simultaneously suppressing CD3+ T cell-mediated adaptive immunity and fostering inflammatory alterations [85,86]. Moreover, *F. nucleatum*, *Enterococcus faecalis*, and *Peptostreptococcus anaerobius* can disrupt the gut barrier, increasing its permeability, thus exposing Toll-like receptors (TLR) on the surface of immune cells in the colon to these bacteria. This process facilitates bacterial invasion, intensifying the inflammatory response. The resultant inflammatory environment produces reactive oxygen species (ROS), leading to DNA damage. This damage may lead to somatic mutations, for example, in one encoding APC, contributing to the development of carcinomas [87].

### 4.4. Metabolism of Dietary Components

Bacterial metabolic byproducts can exert direct carcinogenic effects and contribute to inflammatory changes. Although the mechanisms described in the previous sections influencing the tumorigenesis process are primarily linked to specific bacteria, particularly in food product metabolism, the collective impact of the entire microbial community plays a predominant role [88]. During the fermentation of proteins or amino acids within the colon, bacteria can generate diverse substances, including N-nitroso compounds, polyamines, hydrogen sulfide, and ammonia. These substances give rise to ROS, which promote inflammation or can directly affect DNA, promoting mutations and fostering carcinogenesis [89,90,91]. Another family of compounds whose metabolism by gut bacteria can incite carcinogenesis is bile acids. Upon interaction with intestinal bacteria, the primary bile acids transform into secondary bile acids. The latter induces the generation of ROS and reactive nitrogen species (RNS), subsequently leading to DNA damage. Although only about 5% of primary bile acids typically reach the colon, the excessive consumption of fatty food has been shown to increase that amount [92].

### 4.5. Production of Genotoxins

Genotoxins constitute molecules released by bacteria that can directly damage DNA or alter the activity of tumor suppressor genes. Aside from provoking inflammatory alterations, the BFT secreted by *B. fragilis* serves as a zinc metalloproteinase that dismantles e-cadherin molecules [27]. E-cadherin, an anti-oncogene product, plays a pivotal role in curbing excessive growth signals and facilitating cell–cell adhesion, thereby contributing to the cohesion of epithelial cells. In its absence, epithelial cells become more susceptible to cancer development [27]. Furthermore, *B. fragilis*, as well as *Enterococcus faecalis*, can induce ROS formation that causes direct DNA damage [46,93]. Recent investigations indicate that *Escherichia coli* expressing polyketide synthases (pks+) can synthesize toxins referred to as cyclomodulins, including cytolethal distending toxins (CDT), cytotoxic necrotizing factor (CNF), cycle-inhibiting factor, and colibactin [94]. These toxins exhibit genotoxic properties or interfere with the normal progression of the cell cycle. In addition, enteropathogenic *Escherichia coli* secretes an effector protein called EspF, which promotes carcinogenesis by depleting mismatch repair proteins via posttranscriptional mechanisms and depends on EspF mitochondrial targeting [95].

Figure 1 summarizes the above mechanisms of microbiome alterations impacting colorectal carcinogenesis. 

## 5. Colorectal Cancer and Anti-Neoplastic Medication

The gut microbiome’s influence on antitumor drug therapy primarily occurs via five mechanisms: bacterial translocation, immune regulation, metabolic regulation, enzymatic degradation, and diversity reduction [96]. In this section, we will try to expand on how the response to different systemic therapies for CRC changes with the gut microbiota composition of the host.

5-Fluoro uracil (5-FU) is an antimetabolite primarily used to treat various cancers, especially gastrointestinal cancers, including CRC. It has been shown that 5-FU is associated with a reduction in commensal bacteria such as the genera *Streptococcus* and *Bacteroides* and a concomitant enhancement of Gram-negative bacteria such as *Clostridium hathewayi* and *Lachnospiraceae bacterium* [97]. In their work on mouse models, Atarashi et al. showed that 17 bacterial strains play a crucial role in enhancing Treg cell abundance and inducing anti-inflammatory cytokines, including IL-10. They found that these 17 strains belong to clusters IV, XIVa, and XVIII of Clostridia and that oral administration of these strains to adult mice improved disease in models of colitis [98]. Another study showed that *Lactobacillus plantarum* supernatant (LP SN) amplified the efficacy of 5-FU for CRC and reversed the development of chemoresistance by decreasing cancer stem-like cells [99]. It was shown that the LPSN selectively inhibits the expression of specific markers of CSCs, including CD44, CD133, CD166, and ALDH1. The combination therapy of 5-FU and LP SN leads to increased apoptotic activity by inducing caspase-3 activity, inhibiting the growth of CRCs [100]. Additionally, combination therapy was observed to induce an antitumor mechanism by inactivating the Wnt/Beta-catenin signaling of chemo-resistant cells.

Oxaliplatin, a platinum derivative commonly used in colorectal cancer treatment, also induces immunologic cell death that drives antitumor T-cell immunity [101]. A study completed to understand gut microbiome-related changes due to oxaliplatin showed an increasing Gram-negative bacterial population, with sub-analyses showing an increase in *Prevotella 2* and *Odoribacter* bacteria with a significant reduction in *Prevotella 1* and *Parabacteroides* at the genus level in the oxaliplatin treatment group [102]. In another study performed to evaluate impact of microbiota on the efficacy of oxaliplatin and immunotherapy, mice were inoculated with different tumors, namely EL4 lymphoma, MC38 colon cancer, and B16 melanoma [103]. These mice received antibiotics before tumor inoculation and further throughout the study. The results showed a significantly lower tumor regression and survival in antibiotic-treated mice with EL4 tumors when treated with either oxaliplatin or immunotherapy. There was a downregulation of genes associated with phagocytosis and adaptive immune response along with an upregulation of genes related to tissue development and cancer in antibiotic-treated mice. Additionally, this study reported that antibiotics prevented oxaliplatin-induced DNA damage and apoptotic activity by decreasing ROS and the induction of DNA damage response gene ATR and p53 downstream genes (BAX, FAS, CDKN1A, and RB1) [103].

Irinotecan, also known as CPT-11, is a DNA topoisomerase I inhibitor used to treat CRC. Irinotecan is hydrolyzed by carboxylesterase to form an active metabolite, SN-38. UDP-glucuronyl transferases inactivate SN-38 to its SN-38G form. However, once SN-38G enters the small intestine, it is reactivated by bacterial ꞵ-glucuronidase, leading to several adverse reactions, including diarrhea. Studies have evaluated potent bacterial ꞵ-glucuronidase inhibitors to mitigate CPT-11-induced toxicity, but with mixed results [104,105]

Bevacizumab is a monoclonal antibody against vascular endothelial growth factor A (anti-VEGF A) used in metastatic colorectal cancer and other cancers. In a recent case–control study, a difference in gut microbiome composition was observed between patients with a malignant glioma who received a combination of bevacizumab and temozolomide (group 1) vs. temozolomide monotherapy (group 2). They further found that group 1 patients had a higher abundance of *Firmicutes*, *Bacteroides*, and *Actinobacteria* and a lower abundance of *Bacteroidetes* and *Cyanobacteria* in their fecal microbiota than group 2 patients. Investigations into the potential role of these microbes in modifying treatment response are needed to assess the possibility for fecal microbiota transplants [106].

To conclude, the gut microbiome’s role in influencing chemotherapy effects is increasingly being understood. The supplementation of probiotics and antibiotic treatment to improve the adverse effect profile and prevent chemoresistance has also increasingly been considered, pending further research [96].

Immunotherapy and its association with gut microbiome is another active area of research. Experimental studies in non-CRC cancers, primarily melanoma, have demonstrated differential effects of the gut microbiome on immunotherapy outcomes [107,108]. It was shown that *Bifidobacterium*-treated mice displayed significantly improved tumor control when treated with anti-PD1 therapy compared with their non-*Bifidobacterium*-treated counterparts [107]. Based on a gnotobiotic study involving oral administration of *B. fragilis* to germ-free mice, it was demonstrated that CTLA blockade resistance of colorectal tumors could be overcome [109]. The findings suggest promise for investigating microbiome modulation, such as the adoptive transfer of bacteria-specific T cells or immunization with bacterial products to boost immunotherapy activity [109]. The microbiome could also help to predict immunotherapy-related toxicity based on findings correlating bacterial species and the development of checkpoint-blockade-induced colitis [108].

## 6. Fecal Microbiota Transplantation in Colorectal Cancer

Investigations into the potential relationship between fecal microbiota transplantation (FMT) and colorectal cancer have sparked a new frontier in gastrointestinal research. Emerging studies point to the pivotal role of gut microbiota in influencing the risk and progression of colorectal malignancies. Delving into this intricate connection offers a promising avenue for innovative approaches to combating colorectal cancer. Dysbiosis is associated with cancer and poor outcomes in certain forms of cancer therapy, including allogenic stem cell transplantation [110,111,112,113]. FMT is an intervention by which dysbiosis can be reversed by replacing pathogenic gut microbiomes with microbial species that are more abundant in normal, healthy individuals. Even though FMT is most frequently used in treating *Clostridioides difficile* infection, recent studies show their use in treating other gastrointestinal pathologies like inflammatory bowel disease, hepatic encephalopathy, and colorectal cancer. Therapeutic methods of gut microbiota modification, including FMT and symbiotics, are in the early stages of investigation. Recent research on mouse models has demonstrated a reversal in the microbiome associated with colonic dysplasia post-FMT, with several studies showing near reversal of colonic dysplasia and a significant reduction in pro-oncogenic inflammatory cytokines [114,115,116,117]. These models also show the modulation of immunotherapy efficacy post-FMT. Similar beneficial effects have also been seen in solid tumors other than colorectal cancer [118,119,120]. Regardless, our understanding of this field remains limited, with no similar studies being translated to humans.

For the FMT studies, pathogenic mouse models are created by gavaging azoxymethane (AOM), which is a chemical carcinogen that promotes tumorigenesis, and dextran sodium sulfate (DSS) to disrupt the intestinal barrier and promote colonic inflammation and induces long-lasting colitis. Controls are usually gavaged with isotonic or phosphate-buffered saline (PBS). These mouse models are further gavaged with FMT, either from healthy individuals or from patients with established colorectal cancer. Multiple studies have been conducted with this basic design, showing several unique findings in mice receiving FMT from individuals without CRC compared with pathogenic mice not receiving FMT or receiving FMT from patients with CRC. These findings include the following: (1) A healthier microbiome is associated with a higher percentage of *Firmicutes* than *Bacteroides* [117]. (2) Higher alpha diversity is associated with FMT from control groups without CRC [115,117]. (3) Lesser dysplasia, tumor counts, improved mice weight, and longer intestinal lengths are observed in mice receiving FMT from healthy controls [114,115,117]. (4) Increased inflammatory markers such as IFN-gamma, TNF-alpha, Th1, and Th17 cells, increased Ki-67, PCNA immunostaining, beta-catenin, decreased apoptotic cells are observed in mice receiving a pathogenic gut microbiome [114,115,116]. (5) A decrease in the expression of genes responsible for the intestinal barrier function and colonic immunological barrier can be observed (Table 1) [116]. (6) The inheritance of a higher protection from carcinogens (AOM-DSS) is demonstrated by the offspring of mice receiving FMT from wild population exposed to infections, toxins, and mutagens compared with FMT from lab mice [121]. The last finding emphasizes the importance of considering lineages while selecting the correct donor for interventional studies. The findings are outlined in Figure 2.

Recent studies have hypothesized the role of a healthy microbiome in improving the response to anti-tumor therapy. It has been demonstrated that mice lavaged with feces from patients who responded to immunotherapy as compared with feces from non-responders showed a change in response to immunotherapy post-lavage, offering a potential role of microbiome manipulation in impacting immunotherapy outcomes [118,119,120]. The most widely studied form of immunotherapy involves immune checkpoint inhibiotors (ICI). They have been found to be effective in multiple tumor types [122]. Many studies have shown decreased antitumor effects of ICIs in dysbiotic mouse models treated with antibiotics, rendering them non-responders. Antibiotic-treated mouse models showed a restoration of therapeutic response to ICIs post-fecal microbiota transplantation from responders [118,119,120]. Further results are summarized in Table 1. Surprisingly, these results go beyond chemotherapy and immunotherapy, with one study showing that germ-free mice undergoing total body irradiation combined with FMT before bone marrow transplantation displayed improved survival compared with ones not receiving FMT, suggesting the role of microbiome in preventing radiation-induced toxicity [74].

The overall impact of gut microbiome alterations on colorectal cancer is summarized in Figure 3.

## 7. Conclusions

In CRC patients, certain gut bacteria increase while others decrease, with studies suggesting a potential causative relationship. Even though multiple models have been proposed to understand the microbiome’s oncogenic and cancer-protective properties, the eventual mechanism of action boils down to either the bacterium producing genotoxins or an inflammatory state, either via direct cytokine production or inflammatory metabolite byproducts. Our understanding is constantly improving. Oncotherapy and the gut microbiome show a bidirectional/reciprocal approach rather than a cause–effect relationship. Modalities like chemotherapy, radiotherapy, and immunotherapy alter the gut microbiome drastically. Corollary is also true with changes in the gut microbiome shown to alter the therapeutic response. In such scenarios, FMT emerges as an evolving therapy, showing promise beyond its conventional use in infectious or inflammatory colitis. Even though the mouse studies on FMT and colorectal cancer so far have been promising, the lack of robust human studies remains a notable gap in research.

## Figures and Tables

**Figure 1 microorganisms-12-00484-f001:**
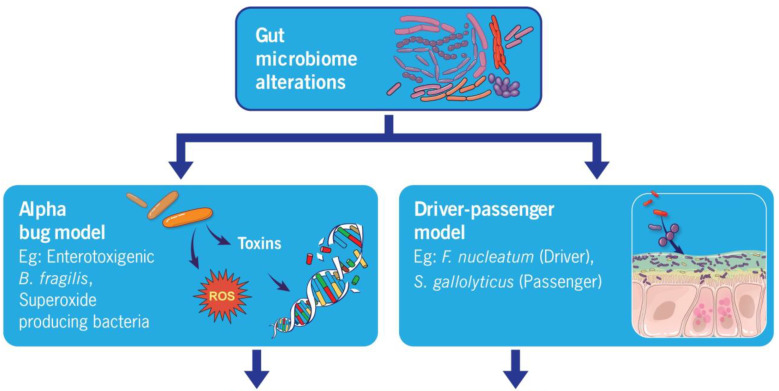
Summarizing different mechanisms of microbiome alterations impacting colorectal carcinogenesis.

**Figure 2 microorganisms-12-00484-f002:**
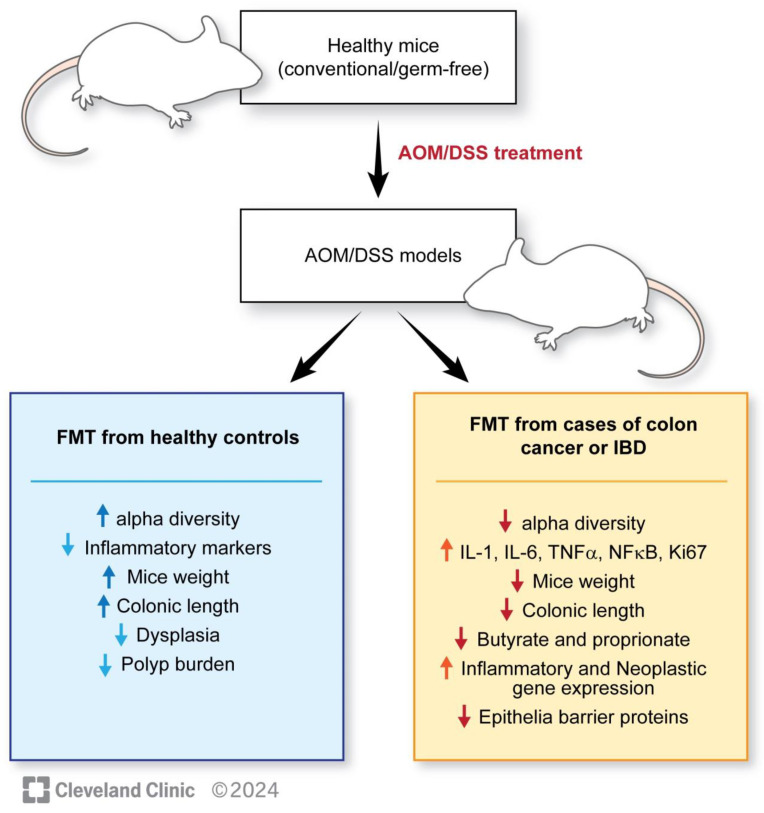
Outlining the results of mice studies related to FMT and colorectal cancer.

**Figure 3 microorganisms-12-00484-f003:**
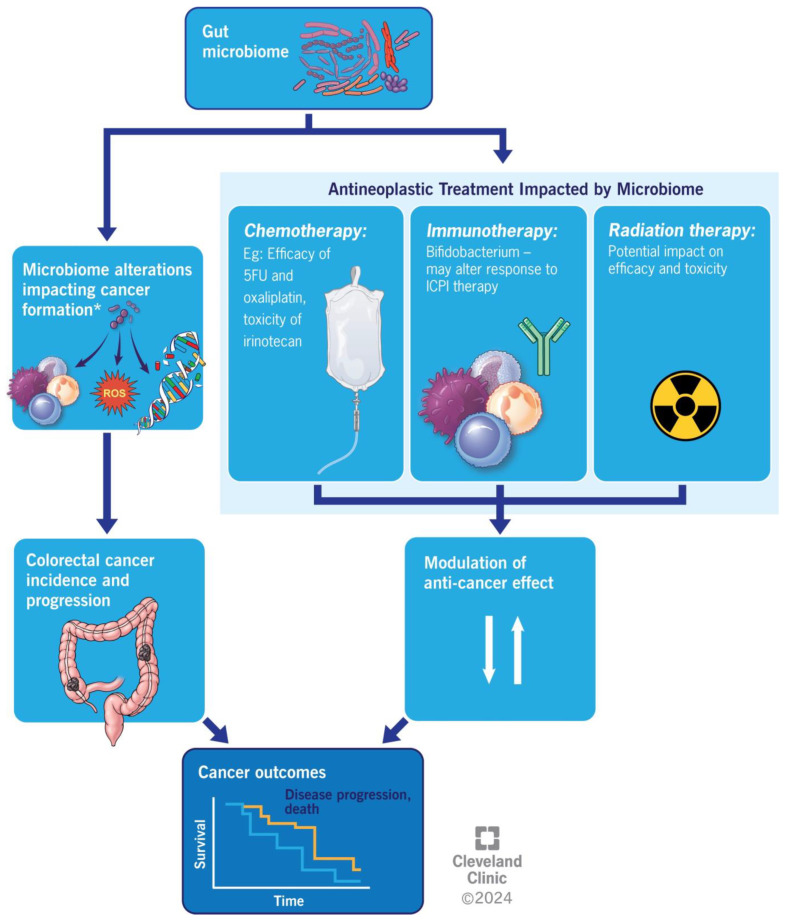
Illustrating the overall impact of gut microbiome alterations on colorectal cancer outcomes.

**Table 1 microorganisms-12-00484-t001:** Review of the literature on mouse model studies showing the efficacy of fecal microbiota transplantation in mice with colorectal cancer and other epithelial tumors.

Study	Design	Results
Wang et al., 2019 [117]	Three mouse groups: CAC+FMT: mice receiving FMT post-AOM-DSS CAC: mice receiving AOM-DSS Control: mice receiving isotonic saline	The CAC group showed an increased percentage of *Bacteroides* as compared with *Firmicutes* when compared with CAC-FMT and the control group. Shannon Index, PD Whole Tree Index and Chao1 Index (measures of alpha diversity) showed a significant increase in the CAC + FMT group. PCA for evaluation of beta diversity in CAC mice showed clustering post-FMT. Decrease in IL-1b, IL-6, TNF-alpha, NF-kB, ki-67 and phospho-p65 levels, increase in IL-10, TGF-beta in CAC + FMT levels at 70 days post-FMT.
Tian et al., 2022 [115]	AOM-treated mice were divided into three groups: FS-HC: AOM mice gavaged with healthy feces FS-UC: AOM mice gavaged with feces from patients with ulcerative colitis (UC) PBs: AOM mice gavaged with phosphate-buffered saline	FS-UC significantly increased the disease activity index, leading to a lower body weight, shorter colon length, a higher number of polyps, more severe dysplasia, a higher Ki-67-positive burden, increased IFN-y, TNF-alpha, Th1 and Th17 expression, and decreased butyrate and propionate concentrations as compared with FS-HC group.
Wong et al., 2017 [114]	The conventional (AOM-treated) and germ-free mouse populations were divided into: CRC-A: conventional mice gavaged with feces of a patient with CRC HC-A: conventional mice gavaged with feces of healthy patients NC-A: conventional mice gavaged with PBS CRC-G: germ-free mice gavaged with feces of patients with CRC HC-G: germ-free mice gavaged with feces of healthy patients	CRC-A group had a higher number of polyps with higher compositive scores, indicative of severe dysplasia and lower bacterial richness. CRC-G group showed increased epithelial proliferation, more Ki-67-positive cells (difference not statistically significant), increased proliferating cell nuclear antigen staining, higher beta-catenin expression, and lower Shannon–Weaver diversity indexes. Both CRC-A and CRC-G groups showed an increase in 33 out of 84 genes associated with inflammation, including Cxcr1, Cxcr2, IL17a, IL22, IL23a, and IFNy-encoding gene. The gene for Tlr-5 was significantly downregulated. Overall, 37 out of the 84 genes involved in cancer pathways also showed upregulation, including Ki-67, Mcm2, Aurka, Cd20, and Bmi1.
Li et al., 2019 [116]	20 C57BL/6 mice and 30 APC gene knockout mice (APC ^min/+^) were used C57BL/6 mice were divided into 2 groups: FMT-CC: gavaged with feces of CRC patients FMT-CH: gavaged with healthy control feces APC ^min/+^ mice were divided into three groups: FMT-AC: gavaged with feces of CRC patients FMT-AH: gavaged with healthy control feces PBS: ones gavaged with PBS	There were no significant changes in mouse, liver, and spleen weight at 8 weeks post-FMT in FMT-AC vs. FMT-AH groups. Overall, 30% of mice in the FMT-AC group showed high-grade dysplasia compared with 10% in the FMT-AH group. Ki-67-positive cells increased in the FMT-AC group. Decrease in ZO-1, occludin, claudin-3, Muc2, cryptdin and Reg3gamma expression and increase in NLRP3, IL-1beta, TNG-alpha and sIgA expression in the small intestine of the FMT-AC group.
Rosshart et al., 2017 [121]	Three categories of mice were selected: Lab: offspring of C57BL/6 mice that did not receive any gavaging. LabR: offspring of germ-free mice that received the frozen gut microbiome from SPF C57BL/6 WildR: offspring of mice receiving the microbiome from wild mice	Results post-AOM-DSS induction in all the categories showed that WildR mice had significantly decreased inflammation, AOM-DSS-induced weight loss, and a lower number and surface area of colorectal tumors with significantly low invasiveness scores compared with Lab and LabR mice.
Routy et al., 2018 [120]	Mice with established MCA-205 sarcoma and RET melanoma were divided into 2 groups. ATB: mice treated with ampicillin + colistin + streptomycin Control: untreated mice Second part of the study: ATB mice inoculated with MCA-205 tumor cells were divided into 2 groups: ATB-R: Receiving FMT from feces of NSCLC responders ATB-NR: Receiving FMT from feces of NSCLC nonresponders	Significantly compromised antitumor effects and survival of ATB mice with PD-1 mAb or in combination with CTLA-4 mAb. Higher microbial richness was associated with the absence of disease progression. Natural progression of sarcoma significantly improved in ATB-R with tumor growth delay, the accumulation of CXCR3 + CD4+ T cells, and the upregulation of PD-L1 in splenic T cells after PD-L1 blockade. *A. mucinophila*, *E. hirae* and *Alistipes indistinctus* from responder stool samples showed restoration of anti-tumor activity of ICIs previously inhibited by antibiotics.
Gopalakrishnan et al., 2018 [118]	Germ-free mouse models injected with BP melanoma cells were divided into 2 parts: R group: receiving gavage from ICI responder patients NR group: receiving gavage from non-responder patients	R group showed significantly decreased tumor size, improved response to anti-PD L1 therapy, increased *Faecalibacterium*, and increased CD45+ and CD8+ T cells. NR group showed increased regulatory CD4+ FoxP3+ T cells and CD4+ IL-17+ T cells in the spleen, suggesting an impaired immune response.
Matson et al., 2018 [119]	Germ-free mice injected with B16.SIY melanoma cells were divided into 3 groups: SPF: GF mice gavaged with feces of Taconic-specific pathogen-free (SPF) mice R group: GF mice gavaged with feces of patients responding to ICI NR group: GF mice gavaged with feces of patients not responding to ICI	SPF mice did not show any change in baseline tumor growth rate. 2 out of 3 mice in the R group showed slow tumor progression compared with 1 out of 3 mice in the NR group.

FMT: fecal michrochromosome maintenance complex component 2; Aurka: auroka kinase A; Bmi1: B-cell-specific Moloney murine leukemia virus integration site 1; APC: adenomatous polyposis coli; ZO-1: zonula occludens-1; Muc2: mucin 2; NLRP3: NOD-like receptor protein 3; TNG-alpha: tumor necrosis factor-alpha; sIgA: secretory immunoglobulin A; SPF: specific pathogen-free; MCA-205: murine colon adenocarcinoma 205; PD-L1: programmed death-ligand 1; ICI: immune checkpoint inhibitor.

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
