# Peer review of "Gut Microbiome–Colorectal Cancer Relationship"

_microorganisms, 2024, doi:10.3390/microorganisms12030484_

Round 1

Reviewer 1 Report

Comments and Suggestions for Authors

 The authors present a review of the relationship between gut microbioma and Colon cancer.

the article is well organised.

Although it is not a new topic, authors tried to collect information on each bacteria of the Microbiome and each new treatment and try to explain its relation with CRC. They also explain the role of micribioma and inflammation as trigger of colorectal carcinogenesis

2 comments

- add the bacterial characteristics of each bacteria

- Did the authors found any direct correlation between the gut Microbiome location and CRC location(ie. right vs left colon?)

Author Response

Thank you for the comments. Please find the attached document communicating the changes in the article. 

Reviewer 2 Report

Comments and Suggestions for Authors

In the manuscript Gut Microbiome-Colon Cancer Relationship the authors summarize the current knowledge about the gut microbiome and its role in the formation and progression of colorectal cancer as well as therapy response. Finally, studies dealing with faecal microbiota transplantation are mentioned. The manuscript is interesting and well written. There are only several points to be corrected before acceptance.

1) Line 76-84 - bigger font of bacterial species

2) Line 94 - different format of citation 1

3) Line 99 - incorrect punctuation

4) Line 107 - dot missing

5) Line 109, 253, 268, 271, 298, 308, 319, 321, 326, 373, 374, 391, 411, Table 1 first and penultimate study - italics missing

6) Line 112 - promotes chemoresistance (without and)?

7) Line 117 - autooxidation

8) Line 117, 230, 314, 326 - extra gap

9) Line 127, 193, 198, 214, 223, 249, 250, 350, 448 - CRC instead of colorectal cancer

10) Line 157, 302, 361, 362 - extra italics

11) Line 258 - extra comma

12) Line 258 - increased instead of Increased

13) Line 286, 291, 347 - introduction of ROS abbreviation was done earlier in the text

14) Line 287 - gap missing 

15) Line 299 - E-cadherin

16) Line 309 - via posttranscriptional ???

17) Line 323 - IL-10 mentioned earlier (introduction of abbreviation)

18) Line 328 - wrong dot location

19) Line 329 - CD133, CD166

20) Line 368, 371 - extra dot

21) Line 371, 397, 413 - please change the format of citation

22) Line 413 - Increased

23) Line 416 - A decrease...

Author Response

Thank you very much for being so thorough with the review of the article. We have addressed all the changes. Please see the document attached below. 

Reviewer 3 Report

Comments and Suggestions for Authors

This review presents a new interesting subject of research in the field of microbiology applied to improving of human health. Authors set the basis of the existing basic science in the use of fecal microbiota transplantation (FMT), gathering the main studies in the bibliography using mice models. In my opinion, this review is an interesting starting point to continue studying with more detail the benefits and potential of FMT in the management of colon cancer.

Although the information is well structured along the paper, I would suggest checking thoroughly the writing construction, since some sentences/paragaphs are difficult to understand.

1.       Check italics between bacteria names (lines 157, 203, 205, 302,)

2.       How do you explain the abundance of Ruminococcus in models of cancer, taking into account that Ruminococcaceae  contribute to structural maintenance and integrity of the gut barrier?

3.       Does statement starting in line 172 “Further analysis showed higher…” refer to previous one (reference 58)?

4.       In line 287-288 bile acids are defined as “dietary component”. I understand involvement on acid biles in digestive process, but I would not refer to them as dietary components.

5.       In line 339 another study is mentioned, but not explained. In line 340 another study is mentioned. Are these two the same study? If so, please rewrite that part. Within this study “EL4 tumors” are mentioned in line 342. If these EL4 are cells, please specify, because as it is written EL4 are undertood to be a type of tumor. Lastly, I do not understand the involvement of oxaliplatin in this study, since it is not mention in the explanation of the study.

6.       In line 412, which subjects refer to characteristics presented in statement number 3?

Comments on the Quality of English Language

This review presents a new interesting subject of research in the field of microbiology applied to improving of human health. Authors set the basis of the existing basic science in the use of fecal microbiota transplantation (FMT), gathering the main studies in the bibliography using mice models. In my opinion, this review is an interesting starting point to continue studying with more detail the benefits and potential of FMT in the management of colon cancer.

Although the information is well structured along the paper, I would suggest checking thoroughly the writing construction, since some sentences/paragaphs are difficult to understand.

1.       Check italics between bacteria names (lines 157, 203, 205, 302,)

2.       How do you explain the abundance of Ruminococcus in models of cancer, taking into account that Ruminococcaceae  contribute to structural maintenance and integrity of the gut barrier?

3.       Does statement starting in line 172 “Further analysis showed higher…” refer to previous one (reference 58)?

4.       In line 287-288 bile acids are defined as “dietary component”. I understand involvement on acid biles in digestive process, but I would not refer to them as dietary components.

5.       In line 339 another study is mentioned, but not explained. In line 340 another study is mentioned. Are these two the same study? If so, please rewrite that part. Within this study “EL4 tumors” are mentioned in line 342. If these EL4 are cells, please specify, because as it is written EL4 are undertood to be a type of tumor. Lastly, I do not understand the involvement of oxaliplatin in this study, since it is not mention in the explanation of the study.

6.       In line 412, which subjects refer to characteristics presented in statement number 3?

Author Response

Thank you for reviewing our manuscript. We have made the changes and are happy to address more if needed. Attaching the document with the changes made. We also submitted the article for English review. 

Round 2

Reviewer 3 Report

Comments and Suggestions for Authors

The authors have satisfactorily answered the questions and the work has been improved. Therefore, I agree that it be accepted in its present form.